# Shared Autonomy with IDA: Interventional Diffusion Assistance

**Brandon J. McMahan**[1], **Zhenghao Peng**[1], **Bolei Zhou**[1], **Jonathan C. Kao**[1]
[1]University of California, Los Angeles
bmcmahan2025@g.ucla.edu   pzh@cs.ucla.edu
bolei@cs.ucla.edu   kao@seas.ucla.edu

## Abstract

The rapid development of artificial intelligence (AI) has unearthed the potential to assist humans in controlling advanced technologies. Shared autonomy (SA) facilitates control by combining inputs from a human pilot and an AI copilot. In prior SA studies, the copilot is constantly active in determining the action played at each time step. This limits human autonomy and may have deleterious effects on performance. In general, the amount of helpful copilot assistance can vary greatly depending on the task dynamics. We therefore hypothesize that human autonomy and SA performance improve through dynamic and selective copilot intervention. To address this, we develop a goal-agnostic intervention assistance (IA) that dynamically shares control by having the copilot intervene only when the expected value of the copilot's action exceeds that of the human's action across all possible goals. We implement IA with a diffusion copilot (termed IDA) trained on expert demonstrations with goal masking. We prove a lower bound on the performance of IA that depends on pilot and copilot performance. Experiments with simulated human pilots show that IDA achieves higher performance than pilot-only and traditional SA control in variants of the Reacher environment and Lunar Lander. We then demonstrate that IDA achieves better control in Lunar Lander with human-in-the-loop experiments. Human participants report greater autonomy with IDA and prefer IDA over pilot-only and traditional SA control. We attribute the success of IDA to preserving human autonomy while simultaneously offering assistance to prevent the human pilot from entering universally bad states.

## 1 Introduction

As technology advances, humans continuously seek to operate more sophisticated and complex devices (Cascio and Montealegre, 2016). However, more sophisticated technologies typically involve complicated operational dynamics and high-dimensional control systems that restrict their use to narrowly defined environments and highly specialized operators (Schulman et al., 2018). While fully autonomous AI agents can be trained to perform these tasks, this approach has three key limitations. First, the user's goal is internalized and not easily deducible in most real-world environments. Second, removing the user from the control loop reduces their autonomy, potentially leading to poor performance and decreased engagement (Wilson and Daugherty, 2018a,b). Third, as the capabilities of AI advance, it is important to consider how to create technologies that assist and empower humans instead of replacing them (Wilson and Daugherty, 2018a,b).

Shared autonomy (SA) addresses these limitations by blending human (pilot) actions with assistive agent (copilot) actions in a closed-loop setting. Prior studies demonstrate SA can increase human task performance in robotic arm control (Laghi et al., 2018), drone control (Reddy et al., 2018), and navigation (Peng et al., 2023). A critical component of prior work is an empirically tuned control-sharing hyperparameter that trades off copilot assistance with human user autonomy (Reddy et al.,

38th Conference on Neural Information Processing Systems (NeurIPS 2024).

Figure 1: Overview of Interventional Assist Framework for control sharing. **(a)** Prior works perform shared autonomy by passing human actions to a copilot (Javdani et al., 2015; Yoneda et al., 2023; Reddy et al., 2018; Jeon et al., 2020). The copilot then plays an action, e.g., by selecting a feasible action closest to the user suggestion (Reddy et al., 2018) or through diffusion (Yoneda et al., 2023). **(b)** In this work, we design an intervention function that plays either the human pilot's action, $a_p$, or the copilot's action, $a_c$, based on their goal-agnostic advantages.

2018; Jeon et al., 2020; Yoneda et al., 2023). Excessive assistance can hinder goal achievement, while insufficient assistance can lead to poor control and performance. Prior work has established multiple methods of control sharing, but limitations remain, such as requiring control sharing hyperparameters be tuned empirically or limiting the copilot to general, non-goal-specific assistance (Schaff and Walter, 2020; Du et al., 2021; Tan et al., 2022). We will discuss them further in Section 2.

Imagine driving a car with an assistive copilot. The majority of the time, the human driver should remain in control. However, the copilot should intervene in certain scenarios to prevent collisions and ensure safe driving. By dynamically adjusting the level of assistance based on the situation and needs of the driver, the copilot simultaneously preserves driver autonomy and engagement while ensuring safety. This example reflects an important open problem: how do we construct an optimal behavior policy from the human and copilot's individual policies? This problem is conceptually similar to the one posed by the Teacher-Student Framework (TSF) (Zimmer et al., 2014), where a teacher agent helps the student learn a good policy by intervening to prevent the student from visiting deleterious states and providing online guidance (Kelly et al., 2019; Li et al., 2022; Peng et al., 2021). A recent work (Xue et al., 2023) has developed methods that improve the TSF framework by using a trajectory-based value estimate to decide when the teacher intervenes in student learning. We propose that a similar value estimate could be used to determine when the copilot should 'intervene' in the user's control, providing the user with assistance. Therefore, we develop an intervention function that estimates the expected value of the copilot and human action in a goal-agnostic fashion. Because our formulation is goal-agnostic, it can generalize to goals not seen during the training process, endowing the human pilot with improved flexibility and generalization.

Our main contribution is a shared control system that leverages a value-based intervention function that can interface with many different copilot architectures while simultaneously improving performance and preserving pilot autonomy. In this work, we build on an existing diffusion-based copilot architecture (Yoneda et al., 2023). These copilots are desirable because they can be trained using supervised methods which helps mitigate the sample complexity required to train effective copilots. While we proceed with the diffusion copilot of (Yoneda et al., 2023) we emphasize that our intervention function can be applied to any copilot capable of generating assistive, alternative, or corrective actions to aid the human in complex tasks.

## 2 Related Work

We have a brief discussion on three related works (Du et al., 2021; Yoneda et al., 2023; Tan et al., 2022).

**Assistance via Empowerment** (Du et al., 2021): It proposes a method that increases a human's ability to control the environment and mitigate the need to infer any goals. It defines an information

theoretic quantity that captures the number of future states accessible to a human from the current state. An assistive agent is then trained to maximize this quantity while the human performs the task. While this allows agents to assist in a goal-agnostic fashion, it typically leads to lower performance than methods that infer the goal, since the assistive agent does not directly help the human achieve a goal. In contrast, other methods, including this work, leverage a copilot that implicitly infers the goal from human actions. While goal inference can lead to lower performance when the goal is incorrectly inferred, we mitigate this by restricting the copilot to only intervene when the human action is deleterious, i.e., worse than the copilot action across all possible goals.

**To the Noise and Back: Diffusion for Shared Autonomy** (Yoneda et al., 2023): It develops a copilot that uses diffusion to map a human action closer to an expert's action. They train a diffusion process to generate actions from a distribution of expert actions conditioned on goal-agnostic state observations. At inference, the human action is first pushed towards a Gaussian distribution by adding noise in the forward diffusion process. The reverse diffusion process is then run on this noised human action to transform it into a sample from the expert action distribution. This action is played in the environment, as illustrated in Figure 1a. The fraction of the forward diffusion process applied to the human action is the diffusion ratio $\gamma \in [0, 1]$ and trades off action conformity (how similar the action is to expert actions) and human autonomy. This copilot therefore requires an experimentally tuned hyperparameter to influence how much control the human has on the eventual action. Because the amount of assistance needed may vary greatly depending on task dynamics, using a fixed amount of assistance throughout the task may limit performance and autonomy. Our intervention function presented in Section 3 addresses this limitation by allowing the copilot to dynamically intervene based on the human's actions.

**On Optimizing Interventions in Shared Autonomy** (Tan et al., 2022): It proposes a method where a copilot is trained with a penalty for intervention. This encourages the copilot to limit its intervention and preserve human autonomy. However, intervention is not inherently bad and need not be sparse. Instead of uniformly penalizing all intervention, only unnecessary intervention should be penalized. Additionally, this method is not inherently hyperparameter free as it does require setting a penalty hyperparameter to determine how the copilot should trade off assistance and autonomy, although they demonstrate this hyperparameter can be solved via optimization. Another limitation is this method assumes access to the human policy during training so that the copilot can learn when best to intervene for a particular human (e.g., an expert at the task would likely have less copilot intervention than a novice). In contrast, we define a general-purpose intervention function based on how good actions are in the environment (irrespective of the specific human pilot playing the actions), enabling one-time training of the intervention function. In our intervention function, an expert would still experience less intervention than a novice because the expert generally plays better actions in the environment. Empirically we find that with our approach the same intervention function can boost the performance of eight different human participants in the Lunar lander environment.

# 3 Method

Our goal is to introduce a general-purpose intervention function that increases the performance and human autonomy of an SA system. The inputs to the intervention function are: (1) the goal-agnostic environment state, $\tilde{s}$, (2) a pilot action, $a_p$, and (3) a copilot action, $a_c$, illustrated in Figure 1b. The intervention function then plays either the pilot action ($a_I = a_p$) or copilot action ($a_I = a_c$) in the environment. We define this intervention function and describe its implementation. We also prove a lower bound on the expected return associated with the policy using the intervention function proportional to the pilot and copilot performance.

We develop an intervention assistance called Interventional Diffusion Assistance (IDA). First, an expert policy is trained to maximize returns in the environment (Section 3.2). Second, the expert is used to perform policy rollouts and generate demonstrations. Goal information is removed from the demonstration data. We then train a diffusion copilot from these demonstrations (Section 3.3). Third, we define a trajectory-based intervention function that decides whether to play the human or copilot action (Section 3.4). All training was performed on a workstation with a single 3080Ti and took approximately 48 hours to complete all three steps for our tasks.

## 3.1 Notation and Problem Formulation.

We assume the environment can be modeled as an infinite-horizon Markov Decision Process (MDP) defined by the tuple $M = \langle S, A, R, \gamma, P, d_0 \rangle$. $S$ is the space of all possible environment states, and $A$ is the space of all possible actions. $R : S \times A \to [R_{\min}, R_{\max}]$ is a scalar reward received for playing an action $a \in A$ in state $s \in S$. $P : S \times A \times S \to [0, 1]$ are the transition dynamics for the environment, $\gamma$ is the discount factor, and $d_0$ is the distribution of initial states. We define the state-action value function induced by policy $\pi$ to be $Q^\pi(s, a) = \mathbb{E}_{s_0=s, a_0=a, a_t \sim \pi(\cdot|s_t), s_{t+1} \sim p(\cdot|s_t, a_t)} \left[ \sum_{t=0}^\infty \gamma^t r(s_t, a_t) \right]$, where $\pi : S \times A \to [0, 1]$ is the action distribution conditioned on the state.

We additionally introduce the notion of a goal that encodes the task objective or intention of the human. We can decompose any state $s = \langle \tilde{s} | \hat{g} \rangle$ into a partial goal-agnostic state observation $\tilde{s}$, which does not contain any goal specific information, and a goal $\hat{g} \in G^*$, where $G^*$ is the space of all possible goals. Then $Q^\pi(s, a) = Q^\pi(\langle \tilde{s} \mid \hat{g} \rangle, a) = Q^\pi(\tilde{s}, a | \hat{g})$ is the state-action value function under the goal-agnostic state $\tilde{s}$ and goal $\hat{g}$.

We model the human behavior by a *human pilot policy*, $\pi_p$, which generates human action $a_p \sim \pi_p(\cdot|s)$ according to the full state observation. The human observes and therefore has access to the goal, $\hat{g} \in G^*$. However, the goal is not assumed to be accessible to the copilot. In this paper, we assume the availability of an *expert policy* $\pi_e(a_e|s)$, that observes the full state and solves the environment. We also assume that we can query the state-action value of the expert policy $Q^{\pi_e}(s, a)$, with which we use to evaluate the quality of an action $a$. We will train a *copilot policy* $\pi_c$ that generates actions based on the pilot action and the goal-agnostic state $a_c \sim \pi_c(\cdot|a_p, \tilde{s})$. The ultimate goal of this paper is to derive a goal-agnostic intervention function $\mathbf{T}(\tilde{s}, a_c, a_p) \in \{0, 1\}$ from the expert policy so that the SA system can achieve better performance than the pilot alone. The behavior policy that shares autonomy between the pilot and the copilot can be represented as $\pi_I = \mathbf{T}\pi_c + (1 - \mathbf{T})\pi_p$.

## 3.2 Training an Expert Policy

We train a soft actor-critic (SAC) expert to solve the environment (Haarnoja et al., 2018) because it allows us to (1) later query $Q^{\pi_e}(s, a)$ for intervention and (2) generate demonstrations in the environment that can be used to train the copilot. In general, other methods of obtaining a $Q$-value estimator and for training a copilot are compatible with IA. We choose SAC for computational convenience. We parameterize our SAC model with a four-layer MLP with 256 units in each layer and the ReLU non-linearity. We use a learning rate of $3 \times 10^{-4}$ and a replay buffer size of $10^6$. The expert fully observes the environment including the goal, and is trained for 3 million time steps or until the environment is solved. We found that training with exploring starts (randomized initial state) produced a more robust $Q$ function with better generalization abilities. Without exploring starts, the $Q$ function performed poorly on unseen states, limiting the effectiveness of IDA.

## 3.3 Training a Diffusion Copilot

Following Yoneda et al. (2023), we trained a diffusion copilot $\pi_c(a_c|a_p, \tilde{s})$ using a denoising diffusion probabilistic model (DDPM) (Ho et al., 2020). For each environment, we collected 10 million state-action pairs from episodes using the SAC expert. All goal information was removed from this demonstration data. The copilot learned to denoise expert actions perturbed with Gaussian noise, conditioned on the goal-agnostic state $\tilde{s}$ and the pilot's action $a_p$.

Formally, the forward diffusion process is a Markov chain that iteratively adds noise $\epsilon \sim \mathcal{N}(0, I)$ according to a noise schedule $\{\alpha_0, \alpha_1, ..., \alpha_T\}$ to an expert action $a_0$, via

$$a_t = \sqrt{a_t} a_{t-1} + \sqrt{1 - \alpha_{t-1}} \epsilon. \tag{1}$$

Following the forward diffusion process, the diffusion copilot is then trained to predict the noise added by the forward process by minimizing the following loss (Ho et al., 2020):

$$\mathcal{L}_{\text{DDPM}} = \mathbb{E}_{t, \tilde{s} \sim \tau, \epsilon \sim \mathcal{N}(0, I)} \left[ \| \epsilon - \epsilon_\theta(a_t, \tilde{s}, t) \|^2 \right], \tag{2}$$

where $\epsilon_\theta$ is a neural network parameterized by $\theta$ that approximates the noise $\epsilon$ conditioned on the noisy action $a_t$, the goal-agnostic state $\tilde{s}$, and the diffusion timestep $t$. $\tau$ is the distribution of states in the demonstration data. The reverse diffusion process is modeled by a four-layer MLP to iteratively refine $a_t$ toward $a_0$.

**Algorithm 1** SA with IDA

---

1: Initialize environment
2: **for** each timestep in episode **do**
3:     Sample state $s$ and goal-masked state $\tilde{s}$ from environment.
4:     Sample human action $a_p \sim \pi_p(\cdot \mid s)$.
5:     Sample diffusion copilot action $a_c \sim \pi_c(\cdot | a_p, \tilde{s})$.
6:     Compute the copilot advantage score $A(\tilde{s}, a_c, a_p)$
7:     **if** $A(\tilde{s}, a_c, a_p) = 1$ **then**
8:         Play copilot action in environment, $a_I = a_c$.
9:     **else**
10:        Play pilot action in environment $a_I = a_p$.
11:     **end if**
12: **end for**
13: environment reset

---

### 3.4 Trajectory-based Goal-agnostic Value Intervention

IDA allows the copilot to intervene in pilot control when they take actions that are consistently bad for all possible goals. We therefore play the copilot's action $a_c$ instead of the pilot's action $a_p$ when the copilot's action has a higher expected return under the expert $Q$-value, that is,

$$Q^{\pi_e}(s, a_c) \geq Q^{\pi_e}(s, a_p). \tag{3}$$

However, we can not directly assess Equation 3, since in a SA system the goal is internal to the pilot. Instead, we only have access to the goal agnostic state $\tilde{s}$, such that, Equation 3 becomes,

$$Q^{\pi_e}(\tilde{s}, a_c) \geq Q^{\pi_e}(\tilde{s}, a_p). \tag{4}$$

We can define an intervention score $\mathbf{I}(\tilde{s}_t, \tilde{a}_t | \hat{g})$ which considers the value of $(\tilde{s}_t, \tilde{a}_t)$ under the assumption that $\hat{g} \in G^*$ is the goal, where $G^*$ is the space of all possible goals,

$$\mathbf{I}(\tilde{s}_t, \tilde{a}_t | \hat{g}) = Q^{\pi_e}(\tilde{s}_t, \tilde{a}_t | \hat{g}). \tag{5}$$

By marginalizing the difference in the intervention scores between the copilot and pilot over the entire goal space we can define a copilot advantage $A(\tilde{s}, a_c, a_p)$

$$\mathbf{A}(\tilde{s}, a_c, a_p) = \frac{1}{G} \int_{\hat{g} \in G^*} F\left(\mathbf{I}(\tilde{s}, a_c | \hat{g}) - \mathbf{I}(\tilde{s}, a_p | \hat{g})\right) d\hat{g}, \tag{6}$$

where $F$ is a function that maps the difference in intervention scores to $\{-1, +1\}$ to ensure all possible goals are weighted equally. Here we choose $F(\cdot) = \text{sign}(\cdot)$. $G$ is a normalization constant for the integral,

$$G = \int_{\hat{g} \in G^*} \max_{\tilde{s}, a_c, a_p} F\left(\mathbf{I}(\tilde{s}, a_c | \hat{g}) - \mathbf{I}(\tilde{s}, a_p | \hat{g})\right) d\hat{g}. \tag{7}$$

When $F(\cdot) = \text{sign}(\cdot)$, then $A(\tilde{s}, a_c, a_p) \in [-1, +1]$ is proportional to the fraction of the goal space over which the copilot action is superior to the pilot action. Also, when $F$ is the sign function, the normalization constant reduces to $G = \int_{\hat{g} \in G^*} d\hat{g}$ and if the goal space is discrete then $G = |G^*|$ is the number of goals.

We adapt the value based intervention function proposed by Xue et al. (2023) to use for shared autonomy by allowing intervention to occur when $A(\tilde{s}, a_c, a_p) = 1$. The copilot therefore intervenes when its action has a higher expected return compared to the pilot action for all possible goals. Formally, we let

$$\mathbf{T}(\tilde{s}, a_c, a_p) = \begin{cases} 1 & \text{if } A(\tilde{s}, a_c, a_p) = 1 \\ 0 & \text{otherwise} \end{cases}, \tag{8}$$

with intervention policy, $\pi_I = \mathbf{T}\pi_c + (1 - \mathbf{T})\pi_p$. The process for performing Shared Autonomy (SA) with IA is highlighted in Algorithm 1. The copilot advantage is computed at every timestep. The behavioral (IA) policy is then determined by Equation 8.

## 3.5 Theoretical Guarantees on the Performance of IA

We prove that the return associated with IA is guaranteed to have the following safety and performance guarantees.

**Theorem 1.** *Let* $J(\pi) = \mathbf{E}_{s_0 \sim d_0, a_t \sim \pi(\cdot|s_t), s_{t+1} \sim P(\cdot|s_t, a_t)}[\sum_{t=0}^{\infty} \gamma^t r(s_t, a_t)]$ *be the expected discounted return of following a policy* $\pi$. *Then, the performance following the Interventional Assistance policy (or behavior policy)* $\pi_I$ *has the following guarantees:*

1. *For a near-optimal pilot,* $(Q^{\pi_e}(s, a_p) \approx \max_{a^*} Q^{\pi_e}(s, a^*))$, $\pi_I$ *is lower bounded by* $\pi_p$:

$$J(\pi_I) \geq J(\pi_p).$$

2. *For a low performing pilot,* $(Q^{\pi_e}(s, a_p) \approx \min_a Q^{\pi_e}(s, a))$, $\pi_I$ *is low bounded by* $\pi_c$:

$$J(\pi_I) \geq J(\pi_c).$$

The proof of Theorem 1 is in Appendix A. Intuitively, the copilot will only intervene when the pilot attempts to play actions from the current state that have expected future returns less than that of the copilot's action across all possible goals. The IA policy therefore does not degrade performance of a high-performing pilot, and when the pilot is poor, guarantees performance no worse than the copilot.

# 4 Results

## 4.1 Experimental Setup

**Baselines.** In the experiments that follow, we compared three different control methods. The first method is pilot-only control. The second method is copilot control, where the behavior policy is equal to the copilot policy $\pi_c(a_c|a_p, \tilde{s})$. Copilot control is the standard practice in SA (Reddy et al., 2018; Yoneda et al., 2023; Schaff and Walter, 2020; Jeon et al., 2020) as it allows a copilot to improve the human action before it is played in the environment. For copilot control, the action played is the action generated by the diffusion copilot using a forward diffusion ratio of $\gamma = 0.2$, the setting that obtained the best control in (Yoneda et al., 2023). Our third method is IDA, which involves dynamically setting the behavior policy based on Equation 8.

**Environments.** The first environment we use is **Reacher**, a 2D simulation environment that models a two-jointed robotic arm with inertial physics. In this environment, torques are applied to the two joints of the robotic arm to position the arm's fingertip at a randomly spawned goal position within the arm's plane of motion. The state of the environment is an 11 dimensional observation containing information about the position and velocities of the joints and goal location. Rewards are given for making smooth trajectories that move the fingertip close to the goal. In each episode, the arm's position is reset to a starting location and a new goal is sampled uniformly across the range of the arm's reach. Following previous works ((Reddy et al., 2018; Schaff and Walter, 2020; Yoneda et al., 2023; Tan et al., 2022)), we also use **Lunar Lander**, a 2D continuous control environment in which a rocket ship must be controlled with three thrusters to land at a desired goal location on the ground. We modify the environment as described in (Yoneda et al., 2023) to make the landing location spawn randomly at different locations along the ground. On each episode the landing zone is indicated by two flags. The states are 9 dimensional observations of the environment containing information about the rocket ship's position, angular velocity, and goal landing zone. We define the success rate as the fraction of episodes that ended with a successful landing between the landing zone flags. We define the crash rate as the fraction of episodes that terminated due to a crash or flying out of bounds.

**Pilots.** We use simulated surrogate pilots (Reacher, Lunar Lander) and eight human pilots (Lunar Lander) to benchmark the performance of pilot-only, copilot, and IDA control (see Appendix D for details about human participants). All human experiments were approved by the IRB and participants were compensated with a gift card. Surrogate control policies are designed to reflect some suboptimalities in human control policies. We consider noisy and laggy surrogate control policies. Surrogate policies are constructed by drawing actions from either an expert policy or a corrupted policy. We use a switch that controls if actions are drawn from the expert or corrupt policies. Actions are initially sampled from the expert policy. At every time step there is a probability of corruption being turned on. Once corruption is turned on, actions are sampled from the corrupt

|            | Continuous |       |       | Linear |       |       | Quadrant |       |       |
|------------|------------|-------|-------|--------|-------|-------|----------|-------|-------|
| Framework  | Expert | Noisy | Laggy | Expert | Noisy | Laggy | Expert | Noisy | Laggy |
| Pilot-Only | 18.8 | 1.6 | 8.5 | 17.7 | 2.07 | 8.94 | 19.05 | 2.07 | 8.38 |
| Copilot    | 0 | 0 | 0 | 0 | 0 | 0 | 0 | 0 | 0.03 |
| IDA (FG)   | 18.0 | 2.9 | 8.5 | 17.3 | 3.1 | 9.0 | 18.5 | 2.9 | 8.4 |
| IDA (DS)   | 18.5 | 2.4 | 8.8 | 18.5 | 2.7 | **9.5** | 17.9 | 2.8 | 8.7 |
| IDA        | **18.8** | **3.3** | **8.7** | **20.0** | **3.1** | 9.4 | **19.7** | **3.6** | **9.2** |

Table 1: Target hit rate (per minute) of surrogate pilots in Reacher environment. "Continuous" uses the default environment goal space where targets spawn anywhere in the arm's plane of motion. "Linear" uses goals that are restricted to a line 100cm in front of the arm. "Quadrant" uses goals that are restricted to the top right quadrant of the workspace. "Copilot" is the pilot with a diffusion copilot (Yoneda et al., 2023) with $\gamma = 0.2$. "IDA" is the pilot with interventional diffusion assistance. IDA (FG) is inferenced with faux goals obtained via monte Carlo sampling and IDA (DS) is inferenced with an expert $Q$ function trained on a different goal distribution of five discrete goals.

control policy. At every time step while corruption is on, there is a probability of turning corruption off. Once corruption is turned off, actions are sampled again from the expert control policy. The noisy surrogate policy is constructed by sampling actions uniformly randomly with a 30% corruption probability. The laggy surrogate policy actions are drawn by repeating the action at the previous time step with an 85% corruption probability.

## 4.2 Reacher Experiments

We compared the performance of IDA to pilot-only and the copilot SA method of (Yoneda et al., 2023) in the Reacher environment with targets that could randomly appear anywhere ("Continuous" in Table 1). We introduce two additional goal spaces to probe the generalization abilities of IDA: "Linear" and "Quadrant." In the linear goal space, goals spawned uniformly random along a horizontal line located 100cm in front of the arm. In the quadrant goal space, goals spawned uniformly random in the upper right quadrant of the workspace. To use IDA without making the goal space known to the advantage computation (Equation 6) we constructed a "faux goal" space (FG) by assuming a uniform distribution over potential next positions as goals. We then estimated the copilot advantage through Monte-Carlo sampling. Furthermore, we examined IDA's performance when the goal space is unknown during $Q$ function training by using a domain shift (DS) environment where goals appear randomly at one of five locations during training. In lieu of using humans, we employed laggy and noisy surrogate control policies to emulate imperfect pilots in the Reacher environment across these goal spaces.

We evaluated performance by quantifying hit rate, the number of targets acquired per minute. In the continuous goal space we found that IDA always achieved performance greater than or equal to pilot-only control and outperformed the copilot (Table 1). The expert control policy was optimal, and IDA therefore rarely intervened with a copilot action, leading to similar performance. We also found the laggy pilot performed relatively well because laggy actions do not significantly impair target reaches, although it may delay target acquisition. When the policy was noisy, IDA improved hit

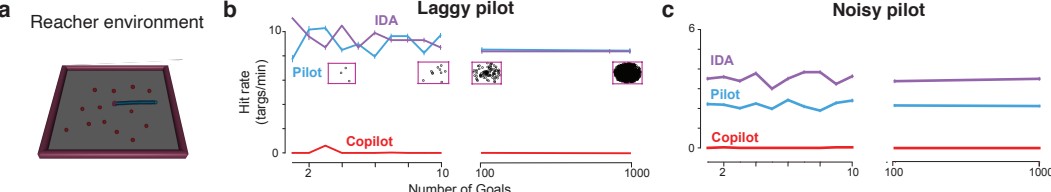

Figure 2: Reacher experiments. **(a)** Continuous Reacher environment. **(b)** Laggy pilot experiments as the number of possible goals varies. IDA performance slightly decreases as the number of possible goals increase, but it never significantly underperforms the pilot. The copilot is significantly worse than both the laggy pilot and IDA. **(c)** Noisy pilot experiments. IDA outperforms the pilot and copilot.

rate from 1.6 to 3.3 targets per minute, approximately doubling performance. In contrast, we found the copilot was unable to acquire any targets. This may be because the copilot is unable to infer the precise goal location from surrogate pilot actions. Together, these results demonstrate that IDA is simultaneously capable of preserving optimal control for high performing pilots while improving hit rate for sub-optimal surrogate control policies, consistent with Theorem 1.

Additionally, we found that even without knowing the goal space during inference (IDA (FG)) or training (IDA (DS)), IDA still increased or maintained the hit rate of the noisy and laggy pilots in the continuous, linear, and quadrant goal spaces (Table 1). We also investigated how the performance of IDA varies with goal space size when goals can appear anywhere (Figure 2a). For each evaluation, we randomly sampled a specified number of candidate goal positions (ranging from 1 to 10, as well as 100 and 1000) from the continuous goal space and then evaluated IDA with goals restricted to these sampled positions. IDA consistently outperformed copilot and the noisy pilot while maintaining the performance of the laggy pilot for all goal space sizes (Figure 2b, c). Collectively, these results demonstrate that IDA performance is robust even when the true goal space is unknown.

## 4.3 Lunar Lander

We next evaluated the performance of IDA in Lunar Lander with the noisy and laggy surrogate control policies. Consistent with Yoneda et al. (2023), we modified Lunar Lander so that the landing zone appears randomly in one of nine different locations. IDA always achieved performance greater than or equal to the performance of the pilot-only control policy (Table 2). Under the expert policy, which achieves 100% success rate and 0% crash rate, IDA does not degrade its performance, although copilot does. We observed that both copilot and IDA improved the performance of noisy and laggy surrogate control policies, with IDA consistently outperforming copilot in successful landings. Copilots and IDA also reduced crash rates for surrogate pilots. Additionally, we compared the performance of IDA to the penalty-based intervention approach proposed by Tan et al. (2022). Because the penalty-based intervention in Tan et al. (2022) used an MLP, we compare it to both IDA and IA with an MLP based copilot. We found that IA consistently achieved a higher rate of successful landings for both the noisy and laggy surrogate pilots than penalty based intervention for both copilot architectures. We further found that IA (MLP and IDA) yielded a lower crash rate than penalty-based intervention.

Next we examined when and why copilot intervention occurred for surrogate pilots. Because these control policies were constructed by corrupting an expert's control policy, we were able to characterize intervention during periods of expert control versus corruption. We examined the distribution of copilot-human advantage scores, which measures the fraction of the goal space over which the copilot's action has a higher expected return than the pilot's action. For both the noisy and laggy pilots, we found the distribution of copilot advantage scores were different during expert actions vs corrupted actions (Figure 3a,b). When corrupted actions were played, there were a greater number of states where the copilot advantage was equal to 1, indicating the copilot's action had a greater expected return over the entire goal space. Consistent with this, we see that intervention was more common during periods of corruption.

| Shared Autonomy | Success Rate | | | Crash Rate | | |
|---|---|---|---|---|---|---|
| | Expert | Noisy | Laggy | Expert | Noisy | Laggy |
| Pilot-only | 100% | 21% | 61% | 0% | 76% | 39% |
| Copilot (MLP) | 92% | 54% | 85% | 0% | 1.6% | 0.7% |
| Copilot (diffusion) | 93.3% | 75% | 92% | 0.3% | 3% | 0.7% |
| Intervention-Penalty (MLP) | **100%** | 58% | 76% | 0% | 4% | 8% |
| IA (MLP) | 99% | **83%** | 96% | 0% | **0.7%** | **0%** |
| IDA (diffusion) | **100%** | **83%** | **100%** | 0% | 3.3% | **0%** |

Table 2: Performance of surrogate pilots in the Lunar Lander environment. IDA is interventional assist with a diffusion copilot. IA (MLP) is interventional assist with an MLP copilot. Intervention-Penalty (MLP) refers to the intervention penalty approach proposed by Tan et al. (2022)

.

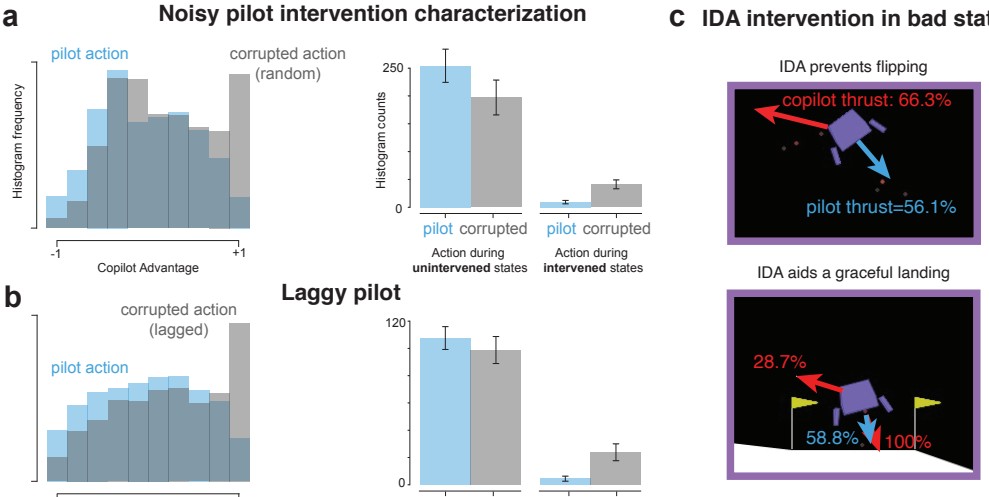

Figure 3: Analysis of copilot advantages during intervened states. **(a)** Characterization of intervention during the noisy pilot control. The left plot shows the copilot advantage, which is generally higher for corrupted (random) actions compared to pilot actions. When quantifying the number of intervened states, we see IDA intervenes more when the corrupted actions are taken. **(b)** Same as **(a)** but for the laggy pilot. **(c)** Example intervened states. In the top panel, the copilot prevents flipping. In the bottom panel, the copilot action helps to make a graceful landing.

We also visualized example states of copilot intervention for both the noisy and laggy pilots (Figure 3c). In two representative examples, we see that intervention generally occurs to stabilize and prevent the rocket ship from crashing or flipping over. In Figure 3c (top), we observe that during intervention, the copilot applies lateral thrust to the rocket to prevent it from rotating sideways despite the pilot operator attempting to fire the main rocket, which would result in a destabilizing rotation of the rocket ship. In Figure 3c (bottom), we observe that intervention occurred during landing where the copilot increased the amount of thrust on the main thruster to soften the touch down while simultaneously applying thrust on the right rocket to level the rocket ship for landing. In both instances, the copilot action prevented the rocket ship from entering a universally low-value state.

### 4.4 Lunar Lander with Human-in-the-loop Control

Given IDA improved the performance of surrogate pilots in Lunar Lander, we performed experiments with eight human participants. Participants played Lunar Lander using pilot-only, copilot, or IDA. Participants used a Logitech game controller with two joysticks to control the rocket ship. The left joystick controlled the lateral thrusters and the right joystick controlled the main thruster. Each participant performed three sequences of 3 experimental blocks (pilot, copilot, IDA) for a total of 9 blocks (see Appendix D for experiment block details). Each block consisted of 30 trials (episodes). Participants were blind to what block they were playing. The game was rendered at 50 fps.

Human pilots experienced considerable difficulty playing Lunar Lander, successfully landing at the goal locations only $14\%$ of the time (Table 3). Copilot control allowed humans participants to successfully land the rocket ship at the goal locations $68.2\%$ of the time. However, the copilot also frequently prevented any landing, resulting in a timeout in $13.8\%$ of trials ($0\%$ in pilot only and $0.1\%$ in IDA). While this led to a lower crash rate ($3.6\%$), it also reduced user autonomy and overall

| Human-in-the-Loop Lunar Lander | | | |
|---|---|---|---|
| | Success Rate | Crash Rate | Timeout Rate | Out of Goal Landing |
| Human Pilot | 14.0 (16.8) % | 85.8 (16.7) % | **0.0** (0.0) | 0.2 (0.6) % |
| w/ Copilot | 68.2 (9.1) % | **3.6** (2.4) % | 13.8 (5.5) | 14.4 (3.6)% |
| w/ IDA | **91.7** (4.9) % | 8.2 (5.1) % | **0.1** (0.4) | **0.0** (0.0) % |

Table 3: Results of human pilots playing Lunar Lander. Mean and (standard error of the mean) presented for each metric.

lowered the success rate. In contrast, IDA achieved the highest performance, enabling participants to successfully land at the goal location $91.7\%$ of the time which was significantly higher than pilot ($p < 0.01$, Wilcoxon signed-rank) and copilot ($p < 0.01$, Wilcoxon signed-rank) control. IDA also resulted in a significant reduction of crash rate when compared to human pilot control ($p < 0.01$, Wilcoxon signed-rank).

To quantify the level of ease, control, and autonomy human participants felt in each block (pilot only, copilot, IDA), we asked participants to provide a subjective rating in response to multiple questions. We asked participants to rate which blocks felt easiest, most controllable, and most autonomous. To assess **ease of control**, we asked: "How easy was the task of landing the rocket ship at the goal location?" To assess **how much control** the participants had, we asked: "How in control did you feel when performing this task?" However, participants may not have felt much control even in pilot-only mode, so to assess **autonomy**, we asked: "How much do you believe your inputs affected the trajectory of the rocketship?" All questions were answered on a scale of 1 to 5 with 5 indicating the easiest, most in control, or most autonomy, respectively. We found that humans subjectively prefer IDA to baseline copilot assistance in terms of ease of use, controllability, and preserving autonomy ($p < 0.01$, Wilcoxon signed-rank).

# 5 Conclusion and Discussion

Our primary contribution is Interventional Assistance (IA): a hyperparameter-free and modular framework that plays a copilot action when it is better than the pilot action across all possible goals. We find that IA outperforms previous methods for intervention based shared autonomy proposed by Tan et al. (2022) as well as traditional copilot-only based methods for control sharing (Yoneda et al., 2023; Reddy et al., 2018). Furthermore, we empirically demonstrated IDA (IA with a **D**iffusion copilot) improves both objective task performance and subjective satisfaction with real human pilots in Lunar Lander (Figure 4). While prior SA systems may degrade pilot performance, particularly when copilots incorrectly infer the pilot's goal (Tan et al., 2022; Du et al., 2021), IA does not degrade human performance (Theorem 1) and often improves it.

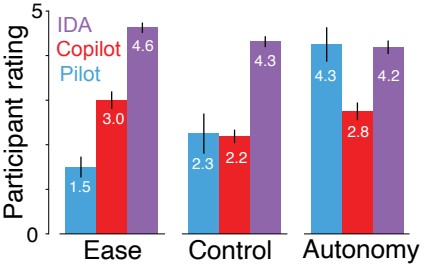

Figure 4: Participants rated IDA as the easiest and most controllable condition. Participants subjectively rated IDA as achieving a similar level of autonomy to pilot only control but significantly better than copilot control.

One limitation of our approach is that we must train an autonomous agent in a simulated environment to obtain an expert $Q$ function. However, this is not a fundamental requirement to learn an intervention function. A straightforward extension of the current work may use an offline dataset of expert demonstrations to train an ensemble of $Q$-networks (Chen et al., 2021). In general, while IA requires access to an expert $Q$ function it makes no assumptions about how that $Q$ function is obtained and we leave various methods of obtaining a $Q$ function as directions for future work.

Additionally, IDA demonstrated resilience across changes in goal space and can be easily adapted to real world settings where the goal space is unknown by construction of these faux goal spaces. Of course, in many settings task structure can be leveraged to further constrain the goal space and improve the assistance IA is able to provide. In these settings, another direction for future work is an implementation of IA that leverages a belief system to differentially weight candidate goals. Future work could potentially improve IA by removing unlikely goals from the advantage computation.

**Acknowledgments**   This project was supported by NSF RI-2339769 (BZ), NSF CPS-2344955 (BZ), the Amazon UCLA Science Hub (JCK), and NIH DP2NS122037 (JCK). ZP is supported by the Amazon Fellowship via UCLA Science Hub.

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

# A   Proof of Theorem 1

In this section, we will prove the theoretical guarantees on the performance of the interventional assistance (IA) in Theorem 1. We first introduce several useful lemmas.

First, we introduce a lemma for later use by following the Theorem 3.2 in (Xue et al., 2023).

**Lemma 1.** *For any behavior policy $\pi_I$ deduced by a copilot policy $\pi_c$, a pilot policy $\pi_p$, and an intervention function $\mathbf{T}(s, a_p, a_c)$, the state distribution discrepancy between $\pi_I$ and $\pi_c$ is bounded by the policy discrepancy and intervention rate:*

$$\|\tau_{\pi_I} - \tau_{\pi_c}\|_1 \leq \frac{(1-\beta)\gamma}{1-\gamma} \mathbb{E}_{s \sim \tau_{\pi_I}} \|\pi_c(\cdot|s) - \pi_p(\cdot|s)\|_1, \tag{9}$$

*where $\beta = \frac{\mathbb{E}_{s \sim \tau_{\pi_I}, a_c \sim \pi_c, a_p \sim \pi_p} \|\mathbf{T}(s, a_p, a_c)[\pi_c(a_c|s) - \pi_p(a_p|s)]\|_1}{\mathbb{E}_{s \sim \tau_{\pi_I}, a_c \sim \pi_c, a_p \sim \pi_p} \|\pi_c(a_c|s) - \pi_p(a_p|s)\|_1}$ is the weighted expected intervention rate. $\tau_{\pi_I}$ and $\tau_{\pi_c}$ are the corresponding state visitation distributions following $\pi_I$ and $\pi_c$, respectively.*

*Proof.* We begin with the result of Theorem 3.2 in (Xue et al., 2023),

$$\begin{aligned}
\|\tau_{\pi_I} - \tau_{\pi_c}\|_1 &\leq \frac{\gamma}{1-\gamma} \mathbb{E}_{s \sim \tau_{\pi_I}} \|\pi_I(\cdot|s) - \pi_c(\cdot|s)\|_1 \\
&= \frac{\gamma}{1-\gamma} \mathbb{E}_{s \sim \tau_{\pi_I}, a_c \sim \pi_c, a_p \sim \pi_p} \|\mathbf{T}\pi_c(a_c|s) + (1-\mathbf{T})\pi_p(a_p|s) - \pi_c(a_c|s)\|_1 \\
&= \frac{\gamma}{1-\gamma} \mathbb{E}_{s \sim \tau_{\pi_I}, a_c \sim \pi_c(\cdot|s), a_p \sim \pi_p(\cdot|s)} \|(1-\mathbf{T}(s, a_p, a_c))[\pi_p(\cdot|s) - \pi_c(\cdot|s)]\|_1 \\
&= \frac{(1-\beta)\gamma}{1-\gamma} \mathbb{E}_{s \sim \tau_{\pi_I}} \|\pi_c(\cdot|s) - \pi_p(\cdot|s)\|_1.
\end{aligned}$$

$\square$

To prove the theorem, the key lemma we use is the policy difference lemma in (Schulman et al., 2015) introduced below. It introduces one policy's advantage function computed on states and actions sampled from another policy's generated trajectory. Here, the advantage function is defined as $A^{\pi'}(s, a) = Q^{\pi'}(s, a) - V^{\pi'}(s)$ and $V^{\pi'}(s) = \mathbb{E}_{a \sim \pi'} Q(s, a)$ is the state value function. $J(\pi) = \mathbb{E}_{s_0 \sim d_0, a_t \sim \pi(\cdot|s_t), s_{t+1} \sim P(\cdot|s_t, a_t)}[\sum_{t=0}^{\infty} \gamma^t r(s_t, a_t)]$ is the expected return following policy $\pi$.

**Lemma 2** (Policy difference lemma). *Let $\pi$ and $\pi'$ be two policies. The difference in expected returns can be represented as follows:*

$$J(\pi) - J(\pi') = \mathbb{E}_{s_t, a_t \sim \tau_\pi} \left[ \sum_{t=0}^{\infty} \gamma^t A^{\pi'}(s_t, a_t) \right] \tag{10}$$

We introduce two lemmas that exploits the intervention function we proposed in Section 3.4.

**Lemma 3.** *The Q value of the behavior action under the expert's Q estimate is greater or equal to the Q value of the pilot action.*

$$\mathbb{E}_{a \sim \pi_I(\cdot|s)} Q^{\pi_e}(s, a) \geq \mathbb{E}_{a_p \sim \pi_p(\cdot|s)} Q^{\pi_e}(s, a_p) \tag{11}$$

*Proof.* According to the intervention function $\mathbf{T}$ in Equation 8, intervention happens when the copilot advantage function $A(\tilde{s}, a_c, a_p) = 1$. If we consider $F$ to be the sign function, then according to Equation 6, $A(\tilde{s}, a_c, a_p) = 1$ means for all goals we will always have $\mathbf{I}(\tilde{s}, a_c|\hat{g}) > \mathbf{I}(\tilde{s}, a_p|\hat{g}), \forall \hat{g}$. Recall $\mathbf{I}(\tilde{s}_t, \tilde{a}_t|\hat{g}) = Q^{\pi_e}(\tilde{s}_t, \tilde{a}_t|\hat{g})$, when intervention happens, we will have:

$$Q^{\pi_e}(\tilde{s}_t, a_c|\hat{g}) > Q^{\pi_e}(\tilde{s}_t, a_p|\hat{g}), \forall \hat{g} \tag{12}$$

Therefore,

$$\mathbf{T}(s, a_c, a_p) Q^{\pi_e}(\tilde{s}_t, a_c|\hat{g}) \geq \mathbf{T}(s, a_c, a_p) Q^{\pi_e}(\tilde{s}_t, a_p|\hat{g}), \forall \hat{g}, \tag{13}$$

where the equality holds when intervention does not happen and $\mathbf{T} = 0$.

Now we introduce the expectation over the behavior policy.

$$\begin{aligned}
\mathbb{E}_{a \sim \pi_I(\cdot|s)} Q^{\pi_e}(\tilde{s}_t, a|\hat{g}) &= \mathbb{E}_{a_p \sim \pi_p} \mathbb{E}_{a_c \sim \pi_c} \mathbf{T} Q^{\pi_e}(\tilde{s}_t, a_c|\hat{g}) + (1-\mathbf{T}) Q^{\pi_e}(\tilde{s}_t, a_p|\hat{g}) \tag{14} \\
&\geq \mathbb{E}_{a_p \sim \pi_p} Q^{\pi_e}(\tilde{s}_t, a_p|\hat{g}), \forall \hat{g} \tag{15}
\end{aligned}$$

The above equation holds for arbitrary $\hat{g}$. Therefore $\mathbb{E}_{a \sim \pi_I(\cdot|s)} Q^{\pi_e}(s, a) \geq \mathbb{E}_{a_p \sim \pi_p} Q^{\pi_e}(s, a_p)$. $\square$

Similar to Lemma 3, we have:

**Lemma 4.** *The Q value of the behavior action under the expert's Q estimate is greater or equal to the Q value of the copilot action.*

$$\mathbb{E}_{a\sim\pi_I(\cdot|s)}Q^{\pi_e}(s,a) \geq \mathbb{E}_{a_c\sim\pi_c}Q^{\pi_e}(s,a_c) \tag{16}$$

*Proof.* According to the intervention function $\mathbf{T}$ in Equation 8, intervention happens when the copilot advantage function $A(\tilde{s},a_c,a_p)=1$. If we consider $F$ to be the sign function, then according to Equation 6, $A(\tilde{s},a_c,a_p)=1$ means for all goals we will always have $\mathbf{I}(\tilde{s},a_c|\hat{g}) > \mathbf{I}(\tilde{s},a_p|\hat{g}), \forall \hat{g}$. Recall $\mathbf{I}(\tilde{s}_t,\tilde{a}_t|\hat{g}) = Q^{\pi_e}(\tilde{s}_t,\tilde{a}_t|\hat{g})$, **when the intervention does not happen**, we will have:

$$Q^{\pi_e}(\tilde{s}_t,a_c|\hat{g}) \leq Q^{\pi_e}(\tilde{s}_t,a_p|\hat{g}), \forall \hat{g} \tag{17}$$

Therefore,

$$(1-\mathbf{T}(s,a_c,a_p))Q^{\pi_e}(\tilde{s}_t,a_p|\hat{g}) \geq (1-\mathbf{T}(s,a_c,a_p))Q^{\pi_e}(\tilde{s}_t,a_c|\hat{g}), \forall \hat{g}, \tag{18}$$

Now we introduce the expectation over the behavior policy.

$$\begin{aligned}
\mathbb{E}_{a\sim\pi_I(\cdot|s)}Q^{\pi_e}(\tilde{s}_t,a|\hat{g}) &= \mathbb{E}_{a_p\sim\pi_p}\mathbb{E}_{a_c\sim\pi_c}\mathbf{T}Q^{\pi_e}(\tilde{s}_t,a_c|\hat{g}) + (1-\mathbf{T})Q^{\pi_e}(\tilde{s}_t,a_p|\hat{g}) &&(19)\\
&\geq \mathbb{E}_{a_c\sim\pi_c}Q^{\pi_e}(\tilde{s}_t,a_c|\hat{g}), \forall \hat{g} &&(20)
\end{aligned}$$

The above equation holds for arbitrary $\hat{g}$. Therefore $\mathbb{E}_{a\sim\pi_I(\cdot|s)}Q^{\pi_e}(s,a) \geq \mathbb{E}_{a_c\sim\pi_c}Q^{\pi_e}(s,a_c)$. $\square$

We introduce another useful lemma.

**Lemma 5** (State Distribution Difference Bound). *Let $\pi$ and $\pi'$ be two policies, and let $\tau_\pi : \mathcal{S} \to [0,1]$ and $\tau_{\pi'} : \mathcal{S} \to [0,1]$ be the corresponding state visitation distributions. For any state-dependent function $f(s)$ that is bounded by $M$ (i.e., $\|f(s)\|_1 \leq M$ for all $s$), the difference in expectations of $f(s)$ under these two distributions is bounded as follows:*

$$\left|\mathbb{E}_{s\sim\tau_\pi}[f(s)] - \mathbb{E}_{s\sim\tau_{\pi'}}[f(s)]\right| \leq M\|\tau_\pi - \tau_{\pi'}\|_1, \tag{21}$$

*where*

$$\|\tau_\pi - \tau_{\pi'}\|_1 = \sum_s |\tau_\pi(s) - \tau_{\pi'}(s)|. \tag{22}$$

*is the total variation distance between two distributions $\tau_\pi$ and $\tau_{\pi'}$.*

*Proof.* The expectation of $f(s)$ under the state distribution $\tau_\pi$ can be written as:

$$\mathbb{E}_{s\sim\tau_\pi}[f(s)] = \sum_s \tau_\pi(s)f(s). \tag{23}$$

Similarly, for policy $\pi'$:

$$\mathbb{E}_{s\sim\tau_{\pi'}}[f(s)] = \sum_s \tau_{\pi'}(s)f(s). \tag{24}$$

The difference in these expectations is:

$$\left|\mathbb{E}_{s\sim\tau_\pi}[f(s)] - \mathbb{E}_{s\sim\tau_{\pi'}}[f(s)]\right| = \left|\sum_s (\tau_\pi(s) - \tau_{\pi'}(s))f(s)\right| \leq \sum_s |\tau_\pi(s) - \tau_{\pi'}(s)|\,|f(s)|. \tag{25}$$

Given that $\|f(s)\|_1 \leq M$, we have:

$$\sum_s |\tau_\pi(s) - \tau_{\pi'}(s)|\,|f(s)| \leq M\sum_s |\tau_\pi(s) - \tau_{\pi'}(s)| = M\|\tau_\pi - \tau_{\pi'}\|_1. \tag{26}$$

Thus, combining these bounds, we have:

$$\left|\mathbb{E}_{s\sim\tau_\pi}[f(s)] - \mathbb{E}_{s\sim\tau_{\pi'}}[f(s)]\right| \leq M\|\tau_\pi - \tau_{\pi'}\|_1. \tag{27}$$

This completes the proof. $\square$

## A.1 The relationship between the behavior policy and the pilot policy

In this section, we will lower bound the performance of the IDA policy $J(\pi_I)$ by the performance of the pilot policy $J(\pi_p)$ under certain conditions.

**Theorem 2.** *Let $J(\pi) = \mathbb{E}_{s_0 \sim d_0, a_t \sim \pi(\cdot|s_t), s_{t+1} \sim P(\cdot|s_t, a_t)}[\sum_{t=0}^{\infty} \gamma^t r(s_t, a_t)]$ be the expected discounted return of following a policy $\pi$. Then for any behavior policy $\pi_I$ deduced by a copilot policy $\pi_c$, a pilot policy $\pi_p$, and an intervention function $\mathbf{T}(s, a_p, a_c)$,*

$$J(\pi_I) \geq J(\pi_p) - \beta R \left[\frac{\gamma}{1-\gamma}\right]^2 \mathbb{E}_{s \sim d_{\pi_I}} \|\pi_c(\cdot \mid s) - \pi_p(\cdot \mid s)\|_1, \tag{28}$$

*wherein $R = R_{max} - R_{min}$ is the range of the reward, $\beta = \frac{\mathbb{E}_{s \sim \tau_{\pi_I}} \|\mathbf{T}(\pi_c(\cdot|s) - \pi_p(\cdot|s))\|_1}{\mathbb{E}_{s \sim \tau_{\pi_I}} \|\pi_c(\cdot|s) - \pi_p(\cdot|s)\|_1}$ is the weighted expected intervention rate, and the $L_1$-norm of output difference $\|\pi_c(\cdot|s) - \pi_p(\cdot|s)\|_1 = \int_{\mathcal{A}} |\pi_c(a|s) - \pi_p(a|s)| \, \mathrm{d}a$ is the discrepancy between $\pi_c$ and $\pi_p$ on state $s$.*

*Proof.* By following Lemma 2, we have

$$J(\pi_I) - J(\pi_e) \tag{29}$$

$$= \mathbb{E}_{s, a \sim \tau_{\pi_I}} \left[\sum_{t=0}^{\infty} \gamma^t A^{\pi_e}(s, a)\right] \tag{30}$$

$$= \mathbb{E}_{s, a \sim \tau_{\pi_I}} \left[\sum_{t=0}^{\infty} \gamma^t \left(Q^{\pi_e}(s, a) - V^{\pi_e}(s)\right)\right] \tag{31}$$

$$= \mathbb{E}_{s \sim \tau_{\pi_I}} \left[\sum_{t=0}^{\infty} \gamma^t \left(\mathbb{E}_{a \sim \pi_I(\cdot|s)} Q^{\pi_e}(s, a) - V^{\pi_e}(s)\right)\right] \tag{32}$$

By following Lemma 3, we continue to have:

$$\mathbb{E}_{s \sim \tau_{\pi_I}} \left[\sum_{t=0}^{\infty} \gamma^t \left(\mathbb{E}_{a \sim \pi_I(\cdot|s)} Q^{\pi_e}(s, a) - V^{\pi_e}(s)\right)\right] \tag{33}$$

$$\geq \mathbb{E}_{s \sim \tau_{\pi_I}} \left[\sum_{t=0}^{\infty} \gamma^t \left(\mathbb{E}_{a_p \sim \pi_p(\cdot|s)} Q^{\pi_e}(s, a_p) - V^{\pi_e}(s)\right)\right] \tag{34}$$

$$= \mathbb{E}_{s \sim \tau_{\pi_I}} \left[\sum_{t=0}^{\infty} \gamma^t \left(\mathbb{E}_{a_p \sim \pi_p(\cdot|s)} A^{\pi_e}(s, a_p)\right)\right] \tag{35}$$

At the same time, we have

$$J(\pi_p) - J(\pi_e) = \mathbb{E}_{s \sim \tau_{\pi_p}} \left[\sum_{t=0}^{\infty} \gamma^t \left(\mathbb{E}_{a_p \sim \pi_p(\cdot|s)} A^{\pi_e}(s, a_p)\right)\right] \tag{36}$$

We want to relate Equation 35 with Equation 36, which has the expectation over state distribution $\tau_{\pi_p}$, instead of $\tau_{\pi_I}$.

To solve the mismatch, we apply Lemma 5. Here, we let

$$f(s) = \sum_{t=0}^{\infty} \gamma^t \mathbb{E}_{a_p \sim \pi_p(\cdot|s)} A^{\pi_e}(s, a_p). \tag{37}$$

We have $\|f(s)\| \leq \frac{\gamma}{1-\gamma} R = M$, where $R = R_{\max} - R_{\min}$ is the range of the reward. Lemma 5 bounds the difference in expectations of $f(s)$ under the state distributions $\tau_{\pi_p}$ and $\tau_{\pi_I}$ as follows:

$$\left|\mathbb{E}_{s \sim \tau_{\pi_p}}[f(s)] - \mathbb{E}_{s \sim \tau_{\pi_I}}[f(s)]\right| \leq M \|\tau_{\pi_p} - \tau_{\pi_I}\|_1 \tag{38}$$

Substituting $f(s)$ into this inequality, we get:

$$\left| \mathbb{E}_{s \sim \tau_{\pi_p}} \left[ \sum_{t=0}^{\infty} \gamma^t \mathbb{E}_{a_p \sim \pi_p(\cdot|s)} A^{\pi_e}(s, a_p) \right] - \mathbb{E}_{s \sim \tau_{\pi_I}} \left[ \sum_{t=0}^{\infty} \gamma^t \mathbb{E}_{a_p \sim \pi_p(\cdot|s)} A^{\pi_e}(s, a_p) \right] \right| \leq \epsilon, \quad (39)$$

where $\epsilon = M \|\tau_{\pi_p} - \tau_{\pi_I}\|_1$.

Thus, we can write:

$$\mathbb{E}_{s \sim \tau_{\pi_I}} \left[ \sum_{t=0}^{\infty} \gamma^t \mathbb{E}_{a_p \sim \pi_p(\cdot|s)} A^{\pi_e}(s, a_p) \right] \geq \mathbb{E}_{s \sim \tau_{\pi_p}} \left[ \sum_{t=0}^{\infty} \gamma^t \mathbb{E}_{a_p \sim \pi_p(\cdot|s)} A^{\pi_e}(s, a_p) \right] - \epsilon, \quad (40)$$

From the deduction above, we have $J(\pi_I) - J(\pi_e) \geq \mathbb{E}_{s \sim \tau_{\pi_I}} \left[ \sum_{t=0}^{\infty} \gamma^t \mathbb{E}_{a_p \sim \pi_p(\cdot|s)} A^{\pi_e}(s, a_p) \right]$ and the right hand side except $\epsilon_c$ is $J(\pi_c) - J(\pi_e)$.

Therefore,

$$J(\pi_I) - J(\pi_e) \geq J(\pi_p) - J(\pi_e) - \epsilon. \quad (41)$$

According to the Theorem 3.2 in (Xue et al., 2023) (Lemma 1), we have

$$\|\tau_{\pi_p} - \tau_{\pi_I}\|_1 \leq \frac{\beta \gamma}{1 - \gamma} \mathbb{E}_{s \sim d_{\pi_I}} \|\pi_c(\cdot \mid s) - \pi_p(\cdot \mid s)\|_1, \quad (42)$$

where $\beta = \frac{\mathbb{E}_{s \sim \tau_{\pi_I}} \|\mathbf{T}(\pi_c(\cdot|s) - \pi_p(\cdot|s))\|_1}{\mathbb{E}_{s \sim \tau_{\pi_I}} \|\pi_c(\cdot|s) - \pi_p(\cdot|s)\|_1}$ is the *weighted expected intervention rate*. Therefore, the $\epsilon$ can be upper bounded by:

$$\epsilon = M \|\tau_{\pi_p} - \tau_{\pi_I}\|_1 \leq \beta R \left[ \frac{\gamma}{1 - \gamma} \right]^2 \mathbb{E}_{s \sim d_{\pi_I}} \|\pi_c(\cdot \mid s) - \pi_p(\cdot \mid s)\|_1. \quad (43)$$

Therefore, we have:

$$J(\pi_I) \geq J(\pi_p) - \beta R \left[ \frac{\gamma}{1 - \gamma} \right]^2 \mathbb{E}_{s \sim d_{\pi_I}} \|\pi_c(\cdot \mid s) - \pi_p(\cdot \mid s)\|_1. \quad (44)$$

This is the lower limit for IDA in terms of the pilot. $\qquad \square$

## A.2 The relationship between the behavior policy and the copilot policy

In this section, we will lower-bound the performance of the IDA policy $J(\pi_I)$ by the performance of the copilot policy $J(\pi_c)$.

**Theorem 3.** *Let* $J(\pi) = \mathbb{E}_{s_0 \sim d_0, a_t \sim \pi(\cdot|s_t), s_{t+1} \sim P(\cdot|s_t, a_t)} [\sum_{t=0}^{\infty} \gamma^t r(s_t, a_t)]$ *be the expected discounted return of following a policy $\pi$. Then for any behavior policy $\pi_I$ deduced by a copilot policy $\pi_c$, a pilot policy $\pi_p$, and an intervention function $\mathbf{T}(s, a_p, a_c)$,*

$$J(\pi_I) \geq J(\pi_c) - (1 - \beta) R \left[ \frac{\gamma}{1 - \gamma} \right]^2 \mathbb{E}_{s \sim d_{\pi_I}} \|\pi_c(\cdot \mid s) - \pi_p(\cdot \mid s)\|_1. \quad (45)$$

*Proof.* According to Lemma 4,

$$J(\pi_I) - J(\pi_e) \quad (46)$$

$$= \mathbb{E}_{s, a \sim \pi_I} \left[ \sum_{t=0}^{\infty} \gamma^t A^{\pi_e}(s, a) \right] \quad (47)$$

$$= \mathbb{E}_{s, a \sim \pi_I} \left[ \sum_{t=0}^{\infty} \gamma^t Q^{\pi_e}(s, a) - V^{\pi_e}(s) \right] \quad (48)$$

$$\geq \mathbb{E}_{s \sim \pi_I} \left[ \mathbb{E}_{a_c \sim \pi_c(\cdot|s)} \sum_{t=0}^{\infty} \gamma^t \left[ Q^{\pi_e}(s, a_c) - V^{\pi_e}(s) \right] \right] \quad (49)$$

With Lemma 2, we have

$$J(\pi_c) - J(\pi_e) = \mathbb{E}_{s \sim \tau_{\pi_c}} \left[ \sum_{t=0}^{\infty} \gamma^t \left( \mathbb{E}_{a_c \sim \pi_c(\cdot|s)} A^{\pi_e}(s, a_c) \right) \right] \tag{50}$$

Now we will use Lemma 5, letting

$$f_c(s) = \sum_{t=0}^{\infty} \gamma^t \mathbb{E}_{a_c \sim \pi_c(\cdot|s)} A^{\pi_e}(s, a_c). \tag{51}$$

We will have $\|f_c(s)\| \leq \dfrac{\gamma}{1-\gamma} R = M$, where $R = R_{\max} - R_{\min}$ is the range of the reward. Lemma 5 bounds the difference in expectations of $f(s)$ under the state distributions $\tau_{\pi_c}$ and $\tau_{\pi_I}$ as follows:

$$\left| \mathbb{E}_{s \sim \tau_{\pi_c}}[f_c(s)] - \mathbb{E}_{s \sim \tau_{\pi_I}}[f_c(s)] \right| \leq M \|\tau_{\pi_c} - \tau_{\pi_I}\|_1 \tag{52}$$

Substituting $f(s)$ into this inequality, we get:

$$\left| \mathbb{E}_{s \sim \tau_{\pi_c}} \left[ \sum_{t=0}^{\infty} \gamma^t \mathbb{E}_{a_c \sim \pi_c(\cdot|s)} A^{\pi_e}(s, a_c) \right] - \mathbb{E}_{s \sim \tau_{\pi_I}} \left[ \sum_{t=0}^{\infty} \gamma^t \mathbb{E}_{a_c \sim \pi_c(\cdot|s)} A^{\pi_e}(s, a_c) \right] \right| \leq \epsilon_c, \tag{53}$$

where $\epsilon_c = M \|\tau_{\pi_c} - \tau_{\pi_I}\|_1$ .

Thus, we can write:

$$\mathbb{E}_{s \sim \tau_{\pi_I}} \left[ \sum_{t=0}^{\infty} \gamma^t \mathbb{E}_{a_c \sim \pi_c(\cdot|s)} A^{\pi_e}(s, a_c) \right] \geq \mathbb{E}_{s \sim \tau_{\pi_c}} \left[ \sum_{t=0}^{\infty} \gamma^t \mathbb{E}_{a_c \sim \pi_c(\cdot|s)} A^{\pi_e}(s, a_c) \right] - \epsilon_c. \tag{54}$$

By substituting Equation 49 and Equation 50 into Equation 54, we have:

$$J(\pi_I) - J(\pi_e) \geq J(\pi_c) - J(\pi_e) - \epsilon_c. \tag{55}$$

By using Lemma 1, we have $\epsilon_c = M \|\tau_{\pi_c} - \tau_{\pi_I}\|_1$ upper-bounded by:

$$\epsilon_c \leq (1 - \beta) R \left[ \frac{\gamma}{1 - \gamma} \right]^2 \mathbb{E}_{s \sim d_{\pi_I}} \|\pi_c(\cdot \mid s) - \pi_p(\cdot \mid s)\|_1 . \tag{56}$$

Therefore, we have:

$$J(\pi_I) \geq J(\pi_c) - (1 - \beta) R \left[ \frac{\gamma}{1 - \gamma} \right]^2 \mathbb{E}_{s \sim d_{\pi_I}} \|\pi_c(\cdot \mid s) - \pi_p(\cdot \mid s)\|_1 . \tag{57}$$

This is the lower bound of IA in terms of the copilot.

$\square$

## A.3 Safety and Performance Guarantees of IA

We now prove our main theorem on the performance guarantees of following the IA policy, restated below:

**Theorem 1.** *Let* $J(\pi) = \mathbb{E}_{s_0 \sim d_0, a_t \sim \pi(\cdot|s_t), s_{t+1} \sim P(\cdot|s_t, a_t)}[\sum_{t=0}^{\infty} \gamma^t r(s_t, a_t)]$ *be the expected discounted return of following a policy* $\pi$. *Then, the performance following the Interventional Assistance policy (or behavior policy)* $\pi_I$ *has the following guarantees:*

1. *For a near-optimal pilot,* $(Q^{\pi_e}(s, a_p) \approx \max_{a^*} Q^{\pi_e}(s, a^*))$, $\pi_I$ *is lower bounded by* $\pi_p$:

$$J(\pi_I) \geq J(\pi_p).$$

2. *For a low performing pilot,* $(Q^{\pi_e}(s, a_p) \approx \min_a Q^{\pi_e}(s, a))$, $\pi_I$ *is low bounded by* $\pi_c$:

$$J(\pi_I) \geq J(\pi_c).$$

*Proof.* From equation 44 and equation 57 we have obtained the following lower bounds on the performance of IA,

$$J(\pi_I) \geq J(\pi_c) - (1 - \beta)R \left[ \frac{\gamma}{1 - \gamma} \right]^2 \mathbb{E}_{s \sim d_{\pi_I}} \| \pi_c(\cdot \mid s) - \pi_p(\cdot \mid s) \|_1 \tag{58}$$

$$J(\pi_I) \geq J(\pi_p) - \beta R \left[ \frac{\gamma}{1 - \gamma} \right]^2 \mathbb{E}_{s \sim d_{\pi_I}} \| \pi_c(\cdot \mid s) - \pi_p(\cdot \mid s) \|_1 \tag{59}$$

For a near optimal pilot,

$$Q(s, a_p) \approx \max_a Q^{\pi_e}(s, a) \tag{60}$$

and therefore,

$$Q(s, a_p) \geq Q(s, a_c). \tag{61}$$

This means that $T(s, a_p, a_c) \approx 0$ and therefore $\beta \approx 0$. When we take the limit of the pilot lower bound as $\beta \to 0$, we have:

$$\lim_{\beta \to 0} \left[ J(\pi_I) \geq J(\pi_p) - \beta R \left[ \frac{\gamma}{1 - \gamma} \right]^2 \mathbb{E}_{s \sim d_{\pi_I}} \| \pi_c(\cdot \mid s) - \pi_p(\cdot \mid s) \|_1 \right] \tag{62}$$

which simplifies to

$$J(\pi_I) \geq J(\pi_p). \tag{63}$$

Similarly for a pilot that is far from optimal,

$$Q(s, a_p) \approx \min_a Q^{\pi_e}(s, a) \tag{64}$$

and therefore

$$Q(s, a_p) \leq Q(s, a_c). \tag{65}$$

This means that $T(s, a_p, a_c) \approx 1$ and therefore $\beta \approx 1$. When we take the limit of the copilot lower bound as $\beta \to 1$, we have:

$$\lim_{\beta \to 1} \left[ J(\pi_I) \geq J(\pi_c) - (1 - \beta)R \left[ \frac{\gamma}{1 - \gamma} \right]^2 \mathbb{E}_{s \sim d_{\pi_I}} \| \pi_c(\cdot \mid s) - \pi_p(\cdot \mid s) \|_1 \right] \tag{66}$$

which simplifies to

$$J(\pi_I) \geq J(\pi_c). \tag{67}$$

We have therefore shown that for a near optimal pilot, IA is guaranteed to not degrade performance, and for a very poor pilot, IA is guaranteed to do at least as good as the copilot. $\square$

## B    Computational Cost of IDA

We found that the computation time for IDA inference only slightly increase as the size of the goal space increases (Table B.1). In all our experiments, we approximate continuous goal spaces by sampling 1,000 candidate goals which costs about 3 ms. This is sufficiently fast for most real-time control applications. Performance could further be improved by using Monte-Carlo estimates as done in the Faux-Goal experiments.

| Time | 1 | 2 | 3 | 4 | 5 | 10 | 100 | 1000 | 10000 | 100000 |
|------|-----|-----|-----|-----|-----|-----|-----|-----|-----|-----|
| (ms) | 2.0 | 2.1 | 2.2 | 2.3 | 2.3 | 2.0 | 2.1 | 3.1 | 2.5 | 11.7 |

Table B.1: Computation time of Advantage Function v. Number of Goals on a single RTX 3080Ti

## C  Examples of Real Human Intervention

We found that, for human experiments in Lunar Lander, interventions by IDA are most common at the start of episodes and when touching down (Figure C.1). Each episode starts with a random disruptive force applied to the rocket so it makes sense that intervention should occur initially to ensure the human maintains a stable flight. Additionally, the rocket ship is very sensitive to the speed and orientation when touching down. Interventions near touch down likely serve to prevent collisions with the ground.

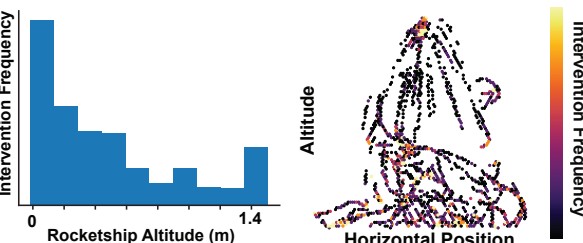

Figure C.1: Intervention tends to occur near the start of trajectories to stabalize the rocket and then again near the end of trajectories to assist the touch down.

## D  Human-in-the-Loop Lunar Lander Experimental Design

Eight (2 female, 6 male) participants with no prior experience performed 3 sequences of Lunar Lander experiments. All participants were compensated with a gift card for their time and involvement in the study. Each sequence consisted of 3 experimental blocks (30 episodes per block) of pilot-only, copilot, and IDA control (Figure D.1). An episode of lunar lander timed out after 30 seconds. In total, each participant played 270 episodes of Lunar Lander, 90 for each condition of pilot-only, copilot, and IDA.

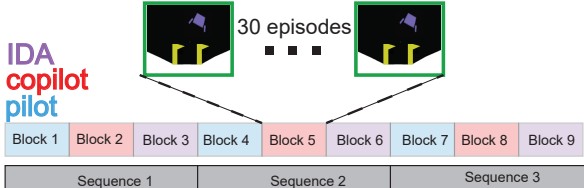

Figure D.1: All participants completed 3 sequences of three blocks. Each sequence was composed of a pilot-only, copilot, and IDA control block. Each block consisted of 30 episodes.

