# OpenReview forum: "Shared Autonomy with IDA: Interventional Diffusion Assistance"
_NeurIPS.cc/2024/Conference — NeurIPS 2024 poster_

### Official Review · Reviewer_zqQF · 2024-07-13

**Soundness:** 3
**Presentation:** 3
**Contribution:** 2
**Rating:** 6
**Confidence:** 3

**Summary:**

In the Shared Autonomy context, the authors propose a value-based intervention assistance method that aims to only have a copilot intervene only when the value of the assistant’s actions exceed that of the human. The proposed method first trains an expert policy for the task using privileged goal information, whose actions are then used to train a diffusion copilot for shared autonomy with the goal information removed. To determine if the copilot's actions should be used instead of the pilot, the method proposes a copilot advantage score, which computes the difference in q-value between the copilot and pilot actions marginalized across all possible goals in the environment. The efficacy of the method is demonstrated with both simulated human pilots and real human-in-the-loop experiments.

**Strengths:**

- To the best of my knowledge, the proposed method is novel and differs sufficiently from relevant prior work with the proposal of the value-based intervention system. Related methods that the paper builds upon are cited, and the experiments compare against the relevant baseline method.
- The experiment setup follows the standard established by similar prior work in deep RL for shared autonomy, including both simulated and human experimenents.
- Generally the submission is well written and clear to read.

**Weaknesses:**

- While Section 2 (Related Works) currently covers more recent relevant work, it would benefit from also providing a more general overview of the shared autonomy literature (e.g. providing an introduction to the shared autonomy and deep RL setting, such as described in "Shared autonomy via deep reinforcement learning" (Reddy et al., 2018) and relevant robotic teleoperation work, such as "A policy-blending formalism for shared control" (Dragan & Srinivasa, 2013)).
- While the proposed method is intuitive, practically it seems difficult to implement since (1) an expert policy needs to be trained beforehand, and (2) the advantage computation at every timestep requires knowledge of the full goal space. A general challenge of the shared autonomy setting is that it is often difficult to capture the full goal space a human has in mind, and it's not clear how the proposed method will work if the agent does not know the full goal space beforehand.
- Having to marginalize out all goals at every timestep for the advantage computation seems quite expensive, especially for higher dimensional goal spaces.
- Figure 3c is quite interesting! it would be strengthened by adding more quantitative evaluation of the findings in Figure 3c (i.e. that the IDA copilot actively intervenes against negative behaviours). For example, computing the dot product between the copilot action and the pilot action vectors to see how much the copilot action differs from the pilot during intervention actions.
- While the user study results show the improvement provided by the method, the sample size of five participants is quite small. I would encourage expanding this sample size and reporting the significance (e.g. p-values) of each of the corresponding metrics (see the similar comparison done in prior work, https://arxiv.org/pdf/1802.01744) .
- Minor comment: renaming $Q^{\pi_e}$ to $\mathbf{I}$ seems unecessary, the former is more clear.

**Questions:**

- Can you clarify what "exploring starts" means for training the SAC expert?
- Have the authors studied how increasing or decreasing the goal space size affects the usability of the method? For example, if we decrease / increase the number of total goals in the reacher environment, are there measurable impacts on the performance of the method? It would be insightful to see a plot showing change in computational efficiency as the number of goals change.

**Limitations:**

The authors do address the limitations of the work. Further discussion on potential societal impact would be helpful (e.g. what are the implications if agents are unable to predict real human goals accurately?)

---

> ### Author Rebuttal · Authors · 2024-08-07
>
> Thank you for your effort to thoroughly review our paper and for your feedback. In response to your feedback, we have made meaningful improvements that have strengthened the study.
>
> > While Section 2 (Related Works) currently covers more recent relevant work, it would benefit from also providing a more general overview of the shared autonomy literature.
>
> We will rewrite text to include a more general overview of shared autonomy.
>
> > While the proposed method is intuitive, practically it seems difficult to implement since (1) an expert policy needs to be trained beforehand, and (2) the advantage computation at every timestep requires knowledge of the full goal space. A general challenge of the shared autonomy setting is that it is often difficult to capture the full goal space a human has in mind, and it's not clear how the proposed method will work if the agent does not know the full goal space beforehand.
>
> We have performed new experiments to show that knowledge of the full goal space **is not required**.  In Reacher experiments, we implemented IDA (FG), using a “Faux Goal” (FG) space where we assume a uniform distribution of goals around the Reacher’s fingertip. We randomly sample goals from this distribution at each inference step. This assumption reflects that in real world scenarios, while the goal locations may be unknown, the user must move the arm in a continuous fashion and therefore any future state near the end effector is a reasonable short term candidate goal. We show that, even without goal space knowledge, IDA improves performance on the Reacher environment for the noisy pilot and maintains the performance of the laggy surrogate control policy. We added this in Table 1 of the rebuttal pdf and will also update the manuscript to include results using this faux goal space.
>
> >Having to marginalize out all goals at every timestep for the advantage computation seems quite expensive, especially for higher dimensional goal spaces. Have the authors studied how increasing or decreasing the goal space size affects the usability of the method? For example, if we decrease / increase the number of total goals in the reacher environment, are there measurable impacts on the performance of the method? It would be insightful to see a plot showing change in computational efficiency as the number of goals change.
>
> We added Table 2 in the rebuttal pdf showing computation time on a single RTX 3080Ti GPU as a function of the goal space size. We find that the computation time only slightly increases as we increase the goal space and it remains under 12ms for goal spaces up to 100,000 candidate goals. This provides up to 80 Hz control, sufficient for most real time experiments. In our experiments we approximate continuous goal spaces by sampling 1,000 candidate goals and we never exceed 1,000. Computation time could be accelerated through using fewer samples, or, as we showed in FG experiments, using Monte-Carlo estimates. We will add this table to the Appendix.
>
> > Figure 3c is quite interesting! it would be strengthened by adding more quantitative evaluation of the findings in Figure 3c (i.e. that the IDA copilot actively intervenes against negative behaviours). For example, computing the dot product between the copilot action and the pilot action vectors to see how much the copilot action differs from the pilot during intervention actions.
>
> Thank you for these suggestions! We have added an additional appendix figure that shows intervention as a function of rocketship altitude and rocketship spatial position (Figure 1 in the rebuttal pdf). This figure shows that interventions typically occur at the very beginning of episodes to stabilize the flight and then increase in frequency at the end of episodes to stabilize the touch down.
>
> > While the user study results show the improvement provided by the method, the sample size of five participants is quite small. I would encourage expanding this sample size and reporting the significance (e.g. p-values) of each of the corresponding metrics (see the similar comparison done in prior work, https://arxiv.org/pdf/1802.01744) .
>
> At the current time we have run additional human experiments and now have a sample size of eight human subjects. We found that humans subjectively prefer IDA to baseline copilot assistance (p < 0.001, Wilcoxon signed-rank) and we also find that humans have a significantly higher success rate with IDA (p < 0.001, Wilcoxon signed-rank) than with baseline copilot control. We have added this additional data along with p-values to our manuscript and figure legends.
>
> > Can you clarify what "exploring starts" means for training the SAC expert?
>
> Exploring starts means that the initial state of the environment is randomized for each training episode, and is a common way to improve exploration in reinforcement learning. We use exploring starts as it improves SAC training. We will clarify this in the manuscript.
>
> > The authors do address the limitations of the work. Further discussion on potential societal impact would be helpful (e.g. what are the implications if agents are unable to predict real human goals accurately?)
>
> IA does not directly infer human goals but instead marginalizes over the space of all possible goals. In cases where no constraints about the goal space are available faux goal spaces can be constructed which essentially just enables IA to prevent universally bad states from occurring. We will add this to our discussion to emphasize that IA can generalize to virtually any real world setting without restricting the users actions.

---

> > ### Comment · Reviewer_zqQF · 2024-08-09
> >
> > I thank the authors for the additional experiments and for answering my questions. In particular, the additional "Faux Goal" experiments and intervention statistics help strengthen the paper. I will be raising my score accordingly.

---

> > > ### Author Response · Authors · 2024-08-10
> > > **Thank-you**
> > >
> > > Thank you again for your attention and thoughtful review of our manuscript. We are grateful for the opportunity to strengthen our manuscript with additional experiments with faux goal space and to more deeply probe when intervention occurs with real human pilots.

---

### Official Review · Reviewer_yfMr · 2024-07-17

**Soundness:** 3
**Presentation:** 3
**Contribution:** 2
**Rating:** 6
**Confidence:** 3

**Summary:**

This work presents the *intervention diffusion assistance* (IDA) framework for shared autonomy between a human "pilot" and an AI "co-pilot".  The IDA framework is designed to be goal agnostic, and does not attempt to infer the pilot's current goal.  This work extends Yoneda et al. (2023), using the same diffusion-based denoising policy, but now only applying the policy in states where the underlying expert policy is believed to be superior to the human pilot's policy.  Their experimental results (with both learned and human pilots) show that this selective denoising policy dominates both the pilot policy and the pilot's policy when corrected using the denoising policy at each step.

References:
1. Yoneda, Takuma, et al. "To the noise and back: Diffusion for shared autonomy." arXiv preprint arXiv:2302.12244 (2023).

**Strengths:**

The main strength of this work is the apparent effectiveness of the selection mechanism that chooses when the co-pilot is allowed to intervene and modify the human pilot's actions.  Experimental results suggest that this mechanism is able to restrict interventions to those states where the co-pilot's policy is most competent, and avoids failures due to "gaps" in the expert's policy.

**Weaknesses:**

One potential weakness with this work is that the intervention selection mechanism depends on knowledge of the goal distribution for the task (as well as a function for "masking" goal information contained in the observations).  Thes would need to be implemented separately for each environment we wish to apply IDA to, and require significant domain expertise.

Another issue is Theorem 1.  The assumptions under which Theorem 1 holds are unclear.  In particular, there appears to be no assumption that the intervention policy is superior to the human policy.  This would seem to be difficult to ensure in general, as the intervention policy is trained over a distribution of goals.  It seems possible that the bounds in Equation 7 would not hold for "low probability" goals.

Additionally, as it was explicitly mentioned that the proposed approach is compatible with any shared autonomy architecture, it would have been nice to see evaluations with the selection mechanism applied to other architectures besides the diffusion-based policy of Yoneda et al.

**Questions:**

1. It was somewhat surprising that in the reacher environment (figure 2) the original denoising policy performs extremely poorly when applied to every state, particularly given its success in the aparently similar box-pushing task in the original work (Yondea et al. 2023).
2. Equation 6 suggests that the advantage must be exactly equal to 1 in order for an intervention to occur, which would seem to be a very rare occurence.  Is this correct, or just a typo?

**Limitations:**

A key limitation of this work that deserves more attention is the need to provide a prior goal distribution, and a mechanism for "masking" goal information in the observations.  Both the prior and the masking function would seem to require significant domain knowledge, and might be difficult to implement in real-world tasks.

---

> ### Author Rebuttal · Authors · 2024-08-07
>
> Thank you for taking the time to carefully read our paper and provide detailed feedback. We have performed additional experiments in response to your invaluable feedback, which we believe has further enhanced and strengthened our work.
>
> > One potential weakness with this work is that the intervention selection mechanism depends on knowledge of the goal distribution for the task (as well as a function for "masking" goal information contained in the observations). Thes would need to be implemented separately for each environment we wish to apply IDA to, and require significant domain expertise.
>
> We first hope to clarify that goal masking and assuming a fixed discrete set of goals is a common practice in the shared autonomy literature (Yoneda et al., Dragon et al., Reddy et al., 2018, Tan et al., 2022). While a recent study (Du et al., 2020) avoids goal inference, it does not perform as well.
> This said, IDA has important advantages over previous work that we hope address your concerns over limiting goal assumptions in the shared autonomy literature:
>
> **IDA works for continuous (infinite) goal spaces.** Although we demonstrated this in our initial submission by testing Reacher with continuous targets, we re-emphasize it here.
>
> **For IDA, the goal space does not need to be known in advance and can be provided at inference.** IDA does not need to know the goal space to be trained. To demonstrate this, we show new experiments where IDA extends to goal spaces it was not trained on. We specifically test a domain shift (DS) environment where the agent was only trained on 5 discrete goals, but generalized to continuous goals.
>
> **IDA works when the goal space is unknown.** To achieve this, we construct a “faux goal” space (FG) by assuming a uniform distribution over potential next positions as goals and achieve a Monte-Carlo estimate of the advantage through sampling. We demonstrate in new experiments that IDA increases performance even in this setting.
>
> We perform additional evaluations of Reacher  (Table 1 in rebuttal pdf) in a continuous goal space, where targets can appear anywhere, as well as goal spaces not observed in training (Linear: goals appear continuously on a line; Quadrant: goals appear in a subset of the space). These results are shown in Table 1 on the attached pdf. IDA increased performance of the noisy pilot in all scenarios, demonstrating that the goal space does not need to be known in advance and that IDA improves performance (1) without a priori knowledge of the goal space (IDA (DS)) and (2) when the goal space is not known (IDA (FG)). We will update the manuscript to include these analyses and include discussion on these advantages in IDA.
>
> Although our experiments show IDA works when the goal space is not prior known and in continuous goal spaces, we do emphasize that in many real world applications, the goal space can be constrained (e.g. location of all graspable objects for a robotic arm with a gripper).
>
> **On goal masking:** In general, when using environments which provide full observations (such as these gym environments) goal masking will be necessary, otherwise the copilot can “cheat” to perform the task goal without inferring it from the user. However, for environments where only partial observations are provided, as is the case in many real world environments, goal masking is no longer necessary.
>
> > Another issue is Theorem 1. The assumptions under which Theorem 1 holds are unclear.
>
> We greatly appreciate your attention to detail here and have decided to restate our theorem and parts of our proof for clarity. In addition to clarifying our theorem, we have revised our theorem to provide two lower bounds on the performance guarantees of IA (instead of just the one that uses the return of the pilot). We have restated the theorem in the attached pdf with assumptions more clearly stated.
>
> > it would have been nice to see evaluations with the selection mechanism applied to other architectures besides the diffusion-based policy of Yoneda et al.
>
> We added additional baselines using an MLP-based copilot that is similar in architecture to Reddy et al., 2018. We have also compared our approach (IA) with the penalty-based intervention proposed by Tan et al., 2022. (Table 3 in rebuttal pdf)
>
> > It was somewhat surprising that in the reacher environment (figure 2) the original denoising policy performs extremely poorly when applied to every state, particularly given its success in the aparently similar box-pushing task in the original work (Yondea et al. 2023).
>
> This performance difference is because the Block Pushing environment used by Yoneda et al., 2023 used only two large discrete goal locations that never changed. We use a continuous distribution of goals in a 2D plane.
>
> > Equation 6 suggests that the advantage must be exactly equal to 1 … Is this correct…?
>
> This is a correct. The advantage is  the fraction of goals where the copilot’s action is superior to the pilot action. Intervention should only occur when the human takes actions that are bad for all possible goals. We will further clarify this in the manuscript.
>
> **Limitations:**
>
> > A key limitation of this work that deserves more attention is the need to provide a prior goal distribution, and a mechanism for "masking" goal information in the observations…  [which] might be difficult to implement in real-world tasks.
>
> To summarize, we have demonstrated that IDA does not require a prior goal distribution. The goal distribution can be changed at inference, and does not even have to be explicitly known (IDA (FG)). Goal masking is necessary in fully observed environments (such as gymnasium) to ensure the copilot does not cheat and infers the user’s goal, but would not be necessary in partially observed environments where the goal is only in the user’s mind (such as reaching to a particular object amongst several).

---

> > ### Comment · Reviewer_yfMr · 2024-08-09
> > **Response to Rebuttals**
> >
> > I believe the authors' responses have addressed my main concerns.  Goal masking is still a potential limitation, but not one that is unique to this work.  I have raised my score accordingly.

---

> > > ### Author Response · Authors · 2024-08-10
> > > **Thank-you**
> > >
> > > Thank you again for your careful review of our manuscript and comments. We are grateful for the opportunity to further discuss goal space considerations and believe additional experiments with a faux goal space has further strengthened our manuscript.

---

### Official Review · Reviewer_Y2yZ · 2024-07-20

**Soundness:** 3
**Presentation:** 3
**Contribution:** 3
**Rating:** 6
**Confidence:** 4

**Summary:**

This paper presents an intervention assistance (IA) method, IDA, that dynamically decides whether the co-pilot should take over the control. The decision is determined by comparing the expected values of the co-pilot’s action versus the pilot’s action. The experiments with human surrogates showed that the proposed IA could improve task performance in the Reacher and Lunar Lander. The human study shows that human subjects prefer IDA.

**Strengths:**

- The proposed method can dynamically adjust whether the copilot policy should intervene to improve ease of use and human autonomy.
- The proposed method follows a goal-agnostic copilot policy, thus can generalize to unseen goal locations.

**Weaknesses:**

- The proposed method still seems to rely on offline datasets to estimate the Q-value. So, the quality of the estimated Q-value is important to the intervention strategy. There is no evidence or discussion on how the performance or impact on users when the expected Q-value is not good enough.
- In the experiments with simulated human surrogates and real humans, the baseline is using the pilot-only or always with a copilot. These two are the most basic baselines. However, since the paper proposes intervention strategies, a better comparison should be the paper such as Tan et al. 2022 where an intervention budget is set. Without such comparison, it is unclear how the proposed intervention advantage is compared with an existing or naive strategy to limit intervention numbers.
   - Weihao Tan, David Koleczek, Siddhant Pradhan, Nicholas Perello, Vivek Chettiar, Vishal Rohra, Aaslesha Rajaram, Soundararajan Srinivasan, H M Sajjad Hossain, and Yash Chandak. On optimizing interventions in shared autonomy, 2022.

**Questions:**

- How is the estimation of Q-value learned in the experiment? Using the same SAC expert or a separate Q-network on the collected data? It is unclear how it is trained and what data it requires.
- If the estimated Q-value is too optimistic or too pessimistic, how does it affect the performance of shared autonomy?
- In Reacher, the laggy action doesn’t affect much about the performance, one explanation can be the domain doesn’t require high-precision control, thus less useful when considering shared control.

**Limitations:**

- The experiments are done with simulated experts, even in real human experiments. In the real world, some domains do not have such simulated/RL experts. How the method transfers or applies to those domains when no simulated/RL is available is not clear.

---

> ### Author Rebuttal · Authors · 2024-08-07
>
> Thank you for taking the time to carefully read through and understand our paper, and provide constructive feedback. We’ve made important changes in response to your feedback (including new experiments) that we believe have significantly improved the manuscript.
>
> > The proposed method still seems to rely on offline datasets to estimate the Q-value. So, the quality of the estimated Q-value is important to the intervention strategy. There is no evidence or discussion on how the performance or impact on users when the expected Q-value is not good enough.
>
> We want to clarify our method does not impose any requirements on how an expert Q function is obtained. In our experiments, it is computationally easy to train a SAC agent in the environment to obtain an expert Q function. However, learning in online simulation or offline data are both equally viable options. We also clarify that IDA depends on obtaining a good Q-value estimator and the performance of IDA will degrade as the quality of the Q-value estimator degrades. We have further incorporated discussion of these factors in the manuscript.
>
> > However, since the paper proposes intervention strategies, a better comparison should be the paper such as Tan et al. 2022 where an intervention budget is set. Without such comparison, it is unclear how the proposed intervention advantage is compared with an existing or naive strategy to limit intervention numbers.
>
> Thank you for raising this point. We performed additional experiments comparing to Tan et al., 2022 (“Intervention-Penalty (MLP)”), as well as Reddy et al., 2018 (“Copilot (MLP”) in the Rebuttal pdf, Table 3, which will be added to the manuscript. We find that IA is always higher performing than the penalty-based assistance proposed by Tan et al., 2022.
> Regarding what comparison is best, we want to highlight some important differences with Tan et al., 2022.
> 1) **IA does not need access to the human control policy.** Penalty-based intervention assumes we can query the human control policy while training the copilot. IA does not.
> 2) **IA better generalizes to continuous action spaces** because the penalty-intervention proposes a penalty whenever the copilot and human actions are different. However, for continuous action spaces the actions are unlikely to ever be equal, necessitating an additional hyper-parameter to decide if the copilot and human actions are sufficiently different to warrant a penalty.
> 3) **IA has no hyperparameters.** The vast majority of previous work in shared autonomy relies on tuning hyper-parameters to optimize control sharing or requiring the copilot to implicitly learn to limit its own assistance (Reddy et al., 2018, Yoneda et al., 2023, Tan et al., 2022, Jeon et al., 2020, Schaff et al., 2020). Tan et al., 2022 requires setting a hyperparameter penalty or budget to determine how the copilot should trade off assistance and autonomy, although it can be solved via optimization.
>
> We will incorporate these new analyses and discussion into the manuscript
>
> > How is the estimation of Q-value learned in the experiment? Using the same SAC expert or a separate Q-network on the collected data? It is unclear how it is trained and what data it requires.
>
> It is learned using the same SAC expert on the collected data. We emphasize our method does not require the Q-value estimator be learned in a particular way. One could also train a Q value from demonstrations collected by a human or heuristic control policy in an online setting or using offline RL methods like Conservative Q Learning [1] or preference-based RL methods like Conservative Preference Learning [2]  to estimate the Q value from the offline data. We chose SAC because it is straightforward in our environments. We will clarify this in the manuscript.
> [1]: Conservative Q-Learning for Offline Reinforcement Learning
> [2]: Contrastive Preference Learning: Learning from Human Feedback without RL
>
> > If the estimated Q-value is too optimistic or too pessimistic, how does it affect the performance of shared autonomy?
>
> IA computes an advantage score that is the difference in Q-values between the copilot and  pilot. Since we consider only relative differences we are resilient to any biases in the Q-value.
>
> > In Reacher, the laggy action doesn’t affect much about the performance, one explanation can be the domain doesn’t require high-precision control, thus less useful when considering shared control.
>
> In Table 1 in our manuscript, the laggy action does degrade performance from 18.8 targets/min to 8.5 targets/min, which is 45% of the initial performance. Given this, we believe the reviewer may have been referring to IDA performance for laggy actions in Reacher. We believe this squares with the intuition that, in Reacher, most of the time laggy actions are not universally bad actions. However, even in this case where IDA offers little assistance we find that IDA is safe to use and does not degrade the laggy pilot’s performance. In Lunar Lander, where laggy actions can be more detrimental, IDA more significantly increases performance.
>
> > The experiments are done with simulated experts, even in real human experiments. In the real world, some domains do not have such simulated/RL experts. How the method transfers or applies to those domains when no simulated/RL is available is not clear.
>
> We wish to clarify that there is no explicit need in our methodology to use simulated experts. Simulated experts were easily trainable for our environments so we made the choice to use them. We follow the methods of Yoneda et al., 2023 which used simulated experts due to computational ease, however, they could have been trained from demonstration data without a simulated RL expert. Depending on the use case, demonstration data collected from real humans, a heuristic control policy, simulated experts, or a combination of the three are all reasonable options for obtaining a Q-value estimator. We will clarify this in the manuscript.

---

> > ### Comment · Reviewer_Y2yZ · 2024-08-09
> >
> > I thank the authors for the response and the additional experiments. They addressed my concerns, I'll raise my score accordingly.

---

> > > ### Author Response · Authors · 2024-08-10
> > > **Thank-you**
> > >
> > > Thank you again for your time and attention in reviewing our manuscript. We are grateful for your comments and the suggestion to compare to Tan et al., 2022 and believe this has lead to a strengthened manuscript.

---

### Author Rebuttal · Authors · 2024-08-07

We thank all the reviewers for their attention and thoughtful comments. We believe this has helped lead to a cleaner and improved manuscript.

We have responded to each of the reviewer's comments individually. Please see the attached pdf for tables and a figure that we refer to in reviewer-specific responses.

We do wish to highlight here that we have performed two additional experiments:

1. We have added baselines from Reddy et al., 2018 and Tan et al., 2022 and compared them to IA in Lunar Lander and shown that IA is always the highest performing.

2. We have conducted additional experiments in the Reacher environment where the goal space is different between training and inference time (domain shift) or unknown at inference time (faux goals).

Both of these additional results will be updated in our manuscript.

---

### Decision · Program_Chairs · 2024-09-25

**Decision:**

Accept (poster)

**Comment:**

This paper introduces a novel and effective method for the value-based intervention system.
The experimental validation confirms the main claims. The submission is clearly written and organized.
Moreover, the human study shows that human subjects prefer the proposed method.